# The Impact of a Digital Health Pathway on Complications Following HIFU Treatment in Prostate Cancer Patients—A Pre- and Postintervention Study

**DOI:** 10.3390/cancers17213484

**Published:** 2025-10-29

**Authors:** Olga Katzendorn, Alessandro Uleri, Michael Baboudjian, Jean-Baptiste Beauval, Harry Toledano, Vincent Bailly, Guillaume Ploussard, Christophe Tollon

**Affiliations:** 1Department of Urology, La Croix du Sud Hospital, 31130 Quint-Fonsegrives, France; 2Department of Urology, Assistance Publique Hôpitaux de Marseille, 13015 Marseille, France; 3Department of Urology, Martigues Hospital, 13698 Martigues, France; 4Department of Urology, Saint Vincent Clinic, 25000 Besançon, France

**Keywords:** focal therapy, HIFU, urinary tract infections, eHealth, digital health

## Abstract

**Simple Summary:**

The use of digital health systems may enhance patient outcomes and its benefits has been demonstrated recently in prostate cancer patients undergoing surgery Focal therapy, such as high-intensity focused ultrasound (HIFU) treatment, of prostate cancer emerged as alternative to surgery in low and intermediate risk prostate cancer. This study analyzed the impact of a digital application on complications and unplanned contacts with the medical team after focal treatment of prostate cancer. We found a decrease in infectious complications as well as unplanned visits after the use of the application. Hence, the implementation of these programs might be beneficial in focal therapy of prostate cancer.

**Abstract:**

**Background/Objectives**: Digital health pathways, including prehabilitation programs, may help reduce complications after urologic procedures. This study assesses the impact of a digital health intervention on infectious complications, urinary retention, and unplanned patient contacts after high-intensity focused ultrasound (HIFU) treatment for prostate cancer. **Methods**: A pre-/post-intervention study design was applied. The intervention consisted of implementing a mobile health pathway via a mobile application integrated into the perioperative management of patients undergoing HIFU treatment for prostate cancer. Urinary complication rates and unplanned patient contacts with the surgical team before and after implementation were compared using the Mann–Whitney U test. **Results**: 58 patients were included in the analysis. Demographic, tumor, and treatment characteristics were comparable between both groups. The post-intervention group showed a lower incidence of symptomatic urinary infections (3 vs. 10; *p* = 0.019) and fewer unplanned visits (4 vs. 10; *p* = 0.047) after the implementation of the mobile application. No significant differences in rates of acute urinary retention and unplanned communication with the surgical team were observed. **Conclusions**: Integration of a digital health pathway was associated with reduced infectious complications and fewer unplanned visits after HIFU treatment. Incorporating such tools into perioperative management may improve patient outcomes.

## 1. Introduction

Over the last years, focal therapy has become a promising alternative to surgery or external radiotherapy in low- or intermediate-risk prostate cancer [1]. This treatment aims to preserve prostate-adjacent tissue and to minimize functional impairments afterwards by selectively ablating clinically significant prostate cancer without necessarily treating the entire gland [1,2]. Different ablative treatments are currently available, with high-intensity focused ultrasound (HIFU) being one of the most commonly applied options [3]. Nevertheless, peri-interventional short-term complications such as urinary tract infections, bleeding, or urinary retention occur in up to 17%, potentially leading to hospital readmission and impacting patient experience [2,4].

In addition to postoperative rehabilitation, prehabilitation programs have become more common in urologic surgery. These preoperative, multimodal interventions aim to optimize patient preparation and to improve postoperative outcomes and satisfaction [5,6]. Prior studies found that prehabilitation can enhance physical performance and even reduce postoperative complications [7,8]. However, the widespread implementation of such programs is hindered by limited healthcare resources or geographic barriers. Digital health, such as smartphone-based mobile health applications, allows the launch of prehabilitation content and a structured perioperative digital pathway in a cost-efficient, effective way for a great number of patients. These platforms permit continuous updates and direct communication with the surgical team by providing perioperative care in a remote manner [8,9].

Although prehabilitation programs and digital health have already been analyzed in different urologic cancer treatments, to our knowledge, no study has yet evaluated the impact of perioperative digital pathways in patients undergoing HIFU therapy.

In this study, we assessed the impact of a digital perioperative pathway on urinary tract infections after focal therapy.

## 2. Materials and Methods

We included patients undergoing HIFU treatment between March 2023 and February 2025 in urologic centers with established expertise in focal therapy. All procedures were performed according to a standardized protocol, which includes preoperative rectal preparation, antibiotic prophylaxis, postoperative bladder catheterization for five days, and a one-month alpha-blocker prescription. The study followed a pre-/post-intervention design, consisting of a first phase before and a second phase after the implementation of a mobile application. In April 2024, in the second phase, the digital health application “BETTY Coaching” (“BETTY Coaching” program; AIMED2 company, Toulouse, France, https://betty.care) was introduced as a supplement to standard care. All patients undergoing HIFU treatment were instructed to download and use the application, and the medical staff received training on its use. As previously described, the application is available for both smartphones and tablets and was designed to enhance patient engagement and facilitate patient outcome reporting. The patient interface consists of sequential screens, intuitive icons, and progress indicators. It includes questionnaires (e.g., follow-up on incontinence, dysuria, and sexual function), interactive perioperative checklists, and educational content through multimedia modules regarding the procedure and the postoperative course—especially explaining the management of the urethral catheter at home, including its handling as well as hygienic and dietary recommendations. Automated notifications and visual progress tracking supported patient engagement. Furthermore, the patient application communicates with the surgeon’s platform, allowing remote review of patient-reported outcomes and daily postoperative monitoring [8]. The follow-up in this study was three months.

The primary endpoint focused on the occurrence of symptomatic urinary infections requiring hospital readmission or antibiotic treatment. Secondary endpoints included other complications such as acute urinary retention and unplanned visits. Outcomes were compared before and after implementation of the digital health pathway.

Patients’ demographic data, tumor characteristics, and the mentioned specific postoperative complications were evaluated. Categorical variables were reported as frequencies and percentages (%), and continuous variables as medians and ranges. The chi-square or Fisher test was performed for categorical variables, and the Mann–Whitney U test for continuous variables in SPSS 22.0 software (SPSS, Inc., Chicago, IL, USA). A significance level of *p* < 0.05 was applied.

## 3. Results

Patients’ demographic data, tumor characteristics, including tumor localization, size, and pathologic features, as well as treatment characteristics, did not differ significantly between the two groups. The incidence of symptomatic urinary infections and unplanned visits was significantly lower in the post-implementation group. No difference was found in acute urinary retention rates and unplanned communication with the surgical team (including phone calls or e-mails) (Table 1).

## 4. Discussion

This study demonstrates that the perioperative implementation of a structured digital health pathway can reduce infectious complications following HIFU treatment. Urinary infection remains a relevant complication after HIFU, even though antibiotic prophylaxis is applied. Recently, a study identified risk factors such as longer postinterventional catheterization or larger ablation volume to be associated with postinterventional urinary infections in HIFU patients [10]. Given that all patients were discharged with a five-day indwelling urine catheter and most were medical laypersons, enhanced education may be crucial. The BETTY coaching application involves preoperative checklists to ensure optimized preoperative preparation as well as hygienic and dietary recommendations to support catheter management at home. Therefore, patients received detailed instructions on how to manage the catheter at home under hygienic conditions, including when and how to change or empty the catheter bags, as well as the importance of maintaining adequate hydration. There is evidence that hygienic interventions, even when performed by healthcare professionals, may reduce catheter-associated infections [11]. Thus, the digital coaching tool may result in comparable effects.

The positive influence of a digital health pathway combining prehabilitation and postoperative digital guidance has already been shown in patients undergoing radical prostatectomy. After the use of the BETTY coaching, complications, hospital stays, unplanned visits, and hospital readmission could be reduced significantly [8]. This is in line with our findings revealing fewer unplanned visits in the digital intervention group. This may reflect enhanced patient confidence and reduced patient anxiety due to intensified preoperative preparation and continuous access to postoperative guidance. Additionally, the presence of fewer complications may lead to lower health service utilization. However, acute urinary retention seems to be unaffected by the digital intervention, suggesting that mechanical mechanisms rather than educational factors are associated with this complication.

Some limitations have to be mentioned. First, this study is a pre-/post-intervention study without randomization, which may lead to a selection bias. Second, both patient groups are relatively small, limiting the validity of statistical analysis. Therefore, a validation in a bigger randomized cohort is necessary. Third, certain possible confounders that might have influenced both the use of the application and the investigated outcomes, such as patients’ educational status, were not available and could therefore not be considered in this analysis.

## 5. Conclusions

In conclusion, this study demonstrates that the use of a peri-interventional digital health pathway, including hygienic education on bladder catheter management, may reduce postoperative infectious complications and unplanned visits; thus, it may improve clinical outcomes in patients undergoing HIFU therapy. The supplementary implementation of these digital programs should be considered in prostate cancer patients receiving focal therapy.

## Figures and Tables

**Table 1 cancers-17-03484-t001:** Patients’, tumor and intervention characteristics and complications.

	Overall Cohort n = 58	Before n = 28	After n = 30	*p*-Value
Patient and cancer features				
Age, years	74 (69–77)	74 (69–76)	74 (69–78)	0.8
PSA, ng/mL	5.9 (4.8–8.3)	5.9 (4.6–7.9)	6.3 (5.0–8.6)	0.3
Previous prostate surgery, n (%)	16 (27.6)	9 (32.1)	7 (23.3)	0.5
Prostate volume on MRI, cc	40 (30–59)	40 (30–50)	40.5 (30–61)	0.9
PIRADS score, n (%)				0.4
3	1 (1.7)	1 (3.6)	0	
4	48 (82.2)	24 (85.7)	24 (80.0)	
5	9 (15.5)	3 (10.7)	6 (20.0)	
Maximal diameter (index lesion), mm	12 (9–14)	12 (8–14)	13 (10–14)	0.4
Index lesion location, n (%)				0.4
Anterior	11 (19.0)	4 (14.3)	7 (23.3)	
Posterior	47 (81.0)	24 (85.7)	23 (76.7)	
				0.1
Apex	20 (34.5)	8 (28.6)	12 (40.0)	
Median	31 (53.4)	14 (50.0)	17 (56.7)	
Base	7 (12.1)	6 (21.4)	1 (3.3)	
Number of positive biopsies, n	9 (5–20)	9 (5–20)	9.5 (4.5–21.5)	0.8
Grade Group, n (%)				0.3
1	13 (22.4)	4 (14.3)	9 (30.0)	
2	41 (70.7)	23 (82.1)	18 (60.0)	
3	4 (6.8)	1 (3.6)	2 (6.6)	
Total tumor length on biopsy, mm	9 (5–20)	9 (5–20)	9.5 (4.5–21.5)	0.8
Treatment extent, n (%):				0.12
Focal	21 (36.2)	13 (46.4)	8 (26.7)	
Zonal	37 (63.8)	15 (53.6)	22 (73.3)	
Treated volume, cc	7.0 (5.9–8.2)	7.1 (4.7–9.1)	6.9 (6.6–7.9)	0.9
**Postoperative outcomes**				
Symptomatic urinary infection, n (%)	13 (22.4)	10 (35.7)	3 (10.0)	0.019
Acute urinary retention, n (%)	7 (12.1)	4 (14.3)	3 (10.0)	0.6
Unplanned visits ^1^, n (%)	14 (24.1)	10 (35.7)	4 (13.3)	0.047
Unplanned contact with the surgical team ^2^, n (%)	11 (19.0)	7 (25.0)	4 (13.3)	0.25

^1^ Including general practitioner, surgeon, and/or emergency department visits. ^2^ Including phone calls and/or email.

## Data Availability

Data will be available upon request.

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
