# Peer review of "The Impact of a Digital Health Pathway on Complications Following HIFU Treatment in Prostate Cancer Patients—A Pre- and Postintervention Study"

_cancers, 2025, doi:10.3390/cancers17213484_

Round 1

Reviewer 1 Report

Comments and Suggestions for Authors

Page 2, Materials and methods.

  • Patients were included between March 2023 and February 2025. The digital health application was introduced in April 2024. It is not clear how and which patients were included. For example, did all patients included since April 2024 use the application? Was it a choice to use it? When patients did not want to use the application, were they excluded or placed in the “before” group? Please explain.
  • Was there a threshold in considering a patient as a user? What if they only filled in a single questionnaire?
  • How long was the period that the postoperative outcomes were measured?
  • Did patients had a TURP before the HIFU procedure? This might influence for example the number of acute urinary retentions.

Page 3, Table 1

  • The percentage of 76.7 in the “Index lesion location, Posterior – Before” group is not right.
  • There is a large difference between the before and after group in
    • Lesion location: apex (respectively, 14.3% and 30.0%)
    • focal vs zonal treatment

Please explain if (and how) these differences could have affected the results?

Author Response

Comment 1: Page 2, Materials and methods.

  • Patients were included between March 2023 and February 2025. The digital health application was introduced in April 2024. It is not clear how and which patients were included. For example, did all patients included since April 2024 use the application? Was it a choice to use it? When patients did not want to use the application, were they excluded or placed in the “before” group? Please explain.

Response 1: We thank the reviewer for this question. After the implementation of the mobile application all patients who underwent HIFU treatment used Betty. We added this information in the manuscript (page 2, materials and methods).

  • Comment 2: Was there a threshold in considering a patient as a user? What if they only filled in a single questionnaire?

Response 2: We did not used a threshold. Patients were considered as users when they downloaded the application and the patient was certified by the surgeon. Adherence rate to all of the educational materials was 85%.

  • Comment 3: How long was the period that the postoperative outcomes were measured?

Response 3: The Follow-up in this study was 3 month. Ongoing trials are going to analyze a longer follow-up period until 2 years. We added the information in the manuscript (page 2, materials and methods).

  • Comment 4: Did patients had a TURP before the HIFU procedure? This might influence for example the number of acute urinary retentions.

Response 4: As described in table 1 a small proportion of patients had undergone prostate surgery before the HIFU treatment. We agree that this might affect functional outcomes after HIFU treatment. However, there was no significant difference between the pre- and postintervention group in terms of prior prostate surgery (p>0.05). Therefore,  the cohort can be considered homogeneous regarding this variable, and we do not expect prior prostate surgery to have affected the study endpoints.

Page 3, Table 1

  • Comment 5: The percentage of 76.7 in the “Index lesion location, Posterior – Before” group is not right.

Response 5: We thank the reviewer for this remark. We corrected this mistake.

  • Comment 6: There is a large difference between the before and after group in
    • Lesion location: apex (respectively, 14.3% and 30.0%)
    • focal vs zonal treatment
  • please explain if (and how) these differences could have affected the results?

Response 6: We thank the reviewer for this important observation. The percentage values reported by the reviewer were incorrect; specifically, apex lesions were 28.6% in the pre-intervention group and 40% in the post-intervention group. Although numerical differences they did not reach statistical significance by Chi-square testing (p > 0.05; see Table 1).

We agree that the primary endpoint of acute urinary retention might be influenced by the treatment extend or lesion location, however this endpoint was comparable between both groups (p>0.05). Therefore, the non-significant difference of treatment extent and lesion location was not discussed. The significant difference in symptomatic urinary infection and unplanned visits are mainly influenced by the perioperative management, therefore we consider it unlikely that the observed, non-significant differences in lesion location or treatment extent affected the reported outcomes.

Reviewer 2 Report

Comments and Suggestions for Authors

Electronic health can help improve perioperative outcomes and overcome organizational constraints related to human resources and health care–related costs. This study assesses the impact of a digital health intervention on outcomes after high-intensity focused ultrasound (HIFU)-treatment for prostate cancer,which has good clinical significance and application prospects. There are several points worth exploring:

1.As mentioned earlier, this application includes questionnaires, surgical checklists, and educational content related to surgery and postoperative courses, particularly explaining how to manage a urinary catheter at home [7]. The digital health application, as the most important application tool in this article, should be described in detail in the article rather than referenced from other articles.

  1. Emphasize that all medical personnel involved in the study (including fixed executive nurses and doctors) received unified training before the start of the research. Ensure that the information and experience obtained by different patients are highly consistent.
  2. Executor log: Used to record any unexpected situations or deviations encountered during the execution process, providing qualitative information for subsequent data analysis. These will affect the observation endpoint of the study.

4.The demographic data, tumor characteristics, and treatment features of the two groups of patients in Table 1 are not comprehensive enough. For example, demographic data should also include education level (which is important for understanding the application), body mass index, accompanying other disease conditions (which directly affect the observation endpoint outcome), etc; Regarding the treatment characteristics, it is necessary to record the surgical treatment time, special circumstances during the surgical process, etc., to avoid bias that may be caused by other factors.

5.Whether to consider including postoperative pain score, IPSS score and other indicators as observation endpoints to evaluate whether the use of this application can help improve the comfort and quality of life of prostate cancer patients after HIFU surgery, and achieve the goal of accelerating postoperative recovery.

Author Response

Electronic health can help improve perioperative outcomes and overcome organizational constraints related to human resources and health care–related costs. This study assesses the impact of a digital health intervention on outcomes after high-intensity focused ultrasound (HIFU)-treatment for prostate cancer,which has good clinical significance and application prospects. There are several points worth exploring:

Comment 1: As mentioned earlier, this application includes questionnaires, surgical checklists, and educational content related to surgery and postoperative courses, particularly explaining how to manage a urinary catheter at home [7]. The digital health application, as the most important application tool in this article, should be described in detail in the article rather than referenced from other articles.

Response 1: We thank the reviewer for the remark. We add some description in the manuscript (page 2, Section materials and methods).

Comment 2: Emphasize that all medical personnel involved in the study (including fixed executive nurses and doctors) received unified training before the start of the research. Ensure that the information and experience obtained by different patients are highly consistent.

Response 2: We included this information in the methods part (page 2, Section materials and methods).

Comment 3: Executor log: Used to record any unexpected situations or deviations encountered during the execution process, providing qualitative information for subsequent data analysis. These will affect the observation endpoint of the study.

Response 3: We thank the reviewer for this important suggestion. The endpoints investigated in this study were assessable in all patients. In future studies with longer follow-ups we are going to consider to evaluate executor log.

Comment 4: The demographic data, tumor characteristics, and treatment features of the two groups of patients in Table 1 are not comprehensive enough. For example, demographic data should also include education level (which is important for understanding the application), body mass index, accompanying other disease conditions (which directly affect the observation endpoint outcome), etc; Regarding the treatment characteristics, it is necessary to record the surgical treatment time, special circumstances during the surgical process, etc., to avoid bias that may be caused by other factors.

Response 4: We thank the reviewer for this important suggestion. Unfortunately, educational level was not available for all of our patients but we might consider to investigate this in future studies.

We agree that additional confounders might influence the investigated endpoints. However, our demographic, tumor and treatment data revealed a homogeneous study population and included all available comprehensive date for this cohort (table 1). This study aimed to evaluate initial impacts of the mobile application on postoperative results after HIFU treatment, rather than to perform a detailed analysis of HIFU complications. Thus, we focused on fundamental tumor and treatment characteristics. To address the reviewer’s concern regarding potential unmentioned bias we added this point as a limitation in the discussion section (page 4).

Comment 5: Whether to consider including postoperative pain score, IPSS score and other indicators as observation endpoints to evaluate whether the use of this application can help improve the comfort and quality of life of prostate cancer patients after HIFU surgery, and achieve the goal of accelerating postoperative recovery.

Response 5: We thank the reviewer for this valuable suggestion. We agree that an evaluation of postoperative pain, IPSS and additional life quality indicators would provide important insight in postoperative recovery after HIFU treatment. However, the present study focused on acute urinary complications and unplanned patient contact as endpoints. Future studies are required to determine whether the use of the application also contributes to improved postoperative comfort, quality of life and recovery.

Reviewer 3 Report

Comments and Suggestions for Authors

This is an interesting and potentially relevant study; however, there are several fundamental issues that must be addressed before the manuscript can be considered for publication.

The abstract is overly vague and oblique. At present, it does not convey essential details of the study design, methods, or key findings. Readers should be able to extract a concise summary of the study and its implications from the abstract alone. This section needs to be rewritten to provide clear, structured, and informative content.

The manuscript does not specify the type of study conducted. It is unclear whether this was prospective or retrospective in nature, or whether it represents a quality improvement (QI)?

The results show a dramatic difference between pre- and post-implementation groups. However, the manuscript does not provide a plausible explanation or underlying mechanism for these findings. As written, this raises concern about potential confounders or unaccounted biases.

Author Response

This is an interesting and potentially relevant study; however, there are several fundamental issues that must be addressed before the manuscript can be considered for publication.

Comment 1: The abstract is overly vague and oblique. At present, it does not convey essential details of the study design, methods, or key findings. Readers should be able to extract a concise summary of the study and its implications from the abstract alone. This section needs to be rewritten to provide clear, structured, and informative content.

Response 1: We acknowledged this and rewrote the abstract including more information.

Comment 2: The manuscript does not specify the type of study conducted. It is unclear whether this was prospective or retrospective in nature, or whether it represents a quality improvement (QI)?

Response 2: We added this information to the materials and methods part (page 2). The study was a pre-/post-intervention study including patients before and after the implementation of a application. Data was collected prospectively an analysis was performed retrospectively.  

Comment3: The results show a dramatic difference between pre- and post-implementation groups. However, the manuscript does not provide a plausible explanation or underlying mechanism for these findings. As written, this raises concern about potential confounders or unaccounted biases

Response 3: We respectfully disagree with the concern regarding unaccounted confounders. The pre- and post-implementation group did not differ significantly in tumor characteristics, basic demographic data and treatment strategies (table 1). The observed significant reduction in symptomatic urinary infections and unplanned visits are discussed in the manuscript, where plausible mechanisms such as potentially impact of hygienic and dietary recommendations and increased patient confidence due to educational content provided by the application are provided (Table 1, Discussion part (page 4)).

Reviewer 4 Report

Comments and Suggestions for Authors

The authors present an important communication on "The impact of a digital health pathway on complications following HIFU-treatment in prostate cancer patients– A pre- and postintervention study".

In order to enhance the quality of the communication, the authors should explain:

1. The authors focus on analyzing the impact of a digital application:
a. What is the educational status or level of the participants?
b. How do they measure usability?
c. Is the impact measured in the ratio of the number of unscheduled visits or based on the severity of prostate cancer: pre- and post-intervention? Or perhaps based on subsequent complications?

2. I suggest including a section to explain the digital application: interfaces, services, menu, etc.

3. What exactly do they mean by "hygiene education?" How is this guaranteed in the application?

Author Response

The authors present an important communication on "The impact of a digital health pathway on complications following HIFU-treatment in prostate cancer patients– A pre- and postintervention study".

In order to enhance the quality of the communication, the authors should explain:

Comment 1: The authors focus on analyzing the impact of a digital application:
a. What is the educational status or level of the participants?

Response 1: We thank the reviewer for this important suggestion. Unfortunately, educational level was not available for all of our patients but we might consider to investigate this in future studies.

      Comment 2: b. How do they measure usability?

Response 2: Usability was evaluated by using the SUS (System Usability Scale), SUS was in this study 85.9/100.

Comment3: c. Is the impact measured in the ratio of the number of unscheduled visits or based on the severity of prostate cancer: pre- and post-intervention? Or perhaps based on subsequent complications?

Response 3: The impact was assessed by comparing the frequency and proportion of outcomes between the pre- and post-intervention groups (i.e., before and after implementation of the mobile application). This is described in detail in the Materials and Methods section (page 2) and summarized in Table 1.

Comment 4: I suggest including a section to explain the digital application: interfaces, services, menu, etc.

Response 4: We acknowledge this suggestion and added more information in the material and methods part (page 2, material and methods).

Comment 5: What exactly do they mean by "hygiene education?" How is this guaranteed in the application?

Response 5: In the application, several articles provided clear and simple explanations on how patients should manage their catheter at home. For example, patients were advised to change to a larger urine bag in a hygienic manner at night, keep the urine bag below bladder level to ensure continuous drainage, empty the bag regularly, wash their hands before any catheter manipulation, perform daily intimate cleaning with water and mild soap, and maintain adequate hydration (at least 1.5 L per day). In response to the reviewer’s comment, we have added a sentence in the Discussion section (page 4, lines 119-122) to further clarify these hygienic instructions.